# Associations between Ocular Biometry, Refractive Error, and Body Characteristics

Veronica Noya-Padin [1,2], Noelia Nores-Palmas [1], Jacobo Garcia-Queiruga [1,2], Maria J. Giraldez [1,2], Hugo Pena-Verdeal [1,2,*] and Eva Yebra-Pimentel [1,2]

1 Applied Physics Department (Optometry Area), Facultade de Óptica e Optometría, Universidade de Santiago de Compostela, 15705 Santiago de Compostela, Spain; veronicanoya.padin@usc.es (V.N.-P.)
2 Optometry Group, Instituto de Investigación Sanitaria Santiago de Compostela (IDIS), 15706 Santiago de Compostela, Spain
* Correspondence: hugo.pena.verdeal@usc.es; Tel.: +34-881-813610

**Abstract:** Myopia is a refractive error widely spread throughout the world, usually related to excessive axial length (AL) of the eye. This elongation could have severe consequences, even leading to blindness. However, AL varies among subjects, and it may be correlated with other anthropometric parameters. The aim of this study was to evaluate the relationships between AL, body height, refractive error, and sex. A total of 72 eyes of 36 myopic participants with a mean age of $11.1 \pm 1.42$ years (ranging from 8 to 14 years) were included in the study. Participants underwent objective refraction by NVision-K5001, AL measurement by Topcon MYAH biometer, and body height measurement. Significant correlations were observed between AL, body height, and spherical equivalent (SE) (Spearman's correlation, all $p \leq 0.016$). When participants were grouped by AL, significant differences were observed for body height and SE, and when grouped by height percentile, significant differences were observed for AL and SE (Kruskal–Wallis test, all $p \leq 0.006$). There was a significant difference in SE, AL, and body height between genders (Mann–Whitney U test, all $p \leq 0.038$). AL relates to the refractive state of the eye and is also influenced by individual anatomical characteristics.

**Keywords:** myopia; axial length; body height





## 1. Introduction

Myopia is a common refractive error that is rapidly increasing in prevalence worldwide [1]. This refractive error is expected to affect 50% of the global population by 2050, which equates to approximately 5 billion people. Implicitly, the number of people with high myopia will also increase, reaching an estimated 1 billion people [1]. The axial length (AL) of the eye is an important factor in determining its refractive error, and in myopia, it is associated with excessive elongation [2]. This implies the degeneration of the structures of the eye, and as a result, high levels of myopia can be associated with severe ocular complications that can even lead to blindness [2]. Therefore, monitoring ocular biometry plays an important role in maintaining eye health, and several strategies have been developed to control its growth [3–5]. These strategies are applied when myopia progression is most pronounced, specifically during childhood and puberty [6]. As with body growth, eye growth is particularly active during these periods [7]. This raises the question of whether ocular and body growth are controlled by a shared mechanism. Furthermore, if this is the case, it also raises the question of whether the refractive error might also be affected by this mechanism. Previous research has suggested that anthropometric measurements may be a potential risk factor for myopia [8–10]. However, the findings are contradictory, with some studies showing no relationship with refraction [11], while others found an association in emmetropes but not in myopes, where ocular elongation accelerated while body height stabilized [12]. In addition to the above, the relationship between refractive error and sex is also a widely debated issue. Some studies have found higher levels of

myopia and AL in boys [13], while others have observed this trend in girls [14,15]. There have also been studies that have found that myopia is not associated with sex [16], or only associated at certain ages [17]. Moreover, some authors discuss possible generational differences and suggest that part of the effect due to sex is an indirect effect influenced by gender-related habits based on social convictions [18]. This lack of consensus calls for further research to determine whether greater body height may be associated with greater AL and higher myopia and whether there is a relationship between these parameters and sex. Understanding the relationships between the parameters mentioned above can be useful in optimizing the mathematical equations used to predict AL in cases where appropriate equipment is not available. Additionally, it prompts the consideration of additional variables. The equation proposed by Morgan et al. [19] incorporates keratometry and refractive error data. Queiros et al. [20] enriched the model by including the age variable. Studying possible disparities based on factors such as sex or height may be useful in bringing a new approach to these calculations and providing a more complete and accurate result. Similarly, understanding the relationship between these parameters may be valuable for optimizing the equations underlying the myopia progression prediction calculators [21].

The objectives of the present study were as follows: (1) to evaluate the correlation between refractive parameters, AL, body height, and age; (2) to evaluate the correlations between refractive parameters, AL, and age based on the sex of the participants; (3) to analyze differences in body height and ocular refraction based on AL; (4) to analyze differences in AL and ocular refraction based on body height percentile; and (5) to analyze the differences in body height, AL, and ocular refraction based on the sex of the participants.

## 2. Materials and Methods

### 2.1. Participants

For the sample size calculation, the software PS Power and Sample Size Calculations v. 3.1.2 (Copyright by William D. Dupont and Walton D. Plummer) was used. The sample calculation was based on AL as the central parameter in the present study. Previous studies suggest that the standard deviation (SD) of AL in Spanish children is distributed with a value of 0.98 mm [22] and that the required increase in AL for a 1.00 D increment in myopia is 0.35 mm [23]. Based on these data, to have 80% power (type II error associated) for a significance level of $\alpha = 0.05$ (type I error associated) with a confidence level of 95% to detect clinical differences in the analyses performed, the minimum number of participants required is 67 for the final analysis.

Recruitment was conducted among myopic children attending the optometry clinic of the center. Subjects were eligible if they were between 8 and 14 years old [24], with myopic spherical refractive error greater than −0.12 D and cylindrical refractive error less than or equal to 1.50 D. Exclusion criteria were as follows: diagnosis of infection or ocular disease at the time of the study, ocular media opacity, diagnosis of corneal ectasia, dry eye disease, chronic, chromosomal, or syndromic disease or the use of medication that could affect the refractive error [25–27]. None of the participants had employed myopia control methods [12].

Written informed consent was obtained from the legal guardians of individual participants prior to inclusion in the study. The research protocol adheres to the tenets of the Declaration of Helsinki, and approval for the study was obtained from the University Bioethics Committee (approval number: USC 04/2022).

### 2.2. Experimental Procedure

During a one-day visit, participants underwent a battery of tests in the following order: anamnesis, objective assessment of refractive status, ocular biometry, and body height measurement.

Objective refraction was performed using the Shin-Nippon NVISION-K 5001 (Rexxam Co., Kagawa, Japan) [28,29]. This device is an open-field autorefractometer, so the fixation target is external and located at a distance. Hence, since accommodation stimulation is avoided, accurate measurements are obtained without the need for cycloplegic instil-

lation [30]. The device measures refractive error in the ranges of $\pm22.00$ D sphere and $\pm10.00$ D cylinder. The refractive power is measured in 0.12 D steps, and the cylindrical axis is measured in $1^\circ$ increments. For the measurements, the vertex distance was always automatically set at 12 mm [28].

Ocular biometry was performed with the Topcon MYAH biometer (Topcon, Tokyo, Japan) [31] device, which measures AL by applying the principle of optical low-coherence interferometry using an 830 nm superluminescent laser diode. The biometry measurement involves six individual interferometric readings, and the displayed final value is obtained from the average of these readings. The device measures biometry in 0.01 mm steps over a range of 15.00 to 38.00 mm [31].

The body height was measured using a tape measure with 1 cm increments attached to the wall. To avoid possible measurement errors, participants were asked to remove their shoes and stand with their knees straight and their head, heels, buttocks, and shoulder blades in contact with the wall surface [12].

### 2.3. Data Processing

2.3.1. Refractive Components

The autorefractometer NVISION-K 5001 displays values for spherical and cylindrical power and the cylindrical axis in the negative spherocylindrical formula. From these data, the spherical equivalent (SE) and the Jackson cross cylinder vectors at $0^\circ$ (J0) and $45^\circ$ (J45) were calculated as follows in Equations (1)–(3), respectively [32]:

$$SE = \text{Sphere power} + (\text{Cylindrical power}/2), \tag{1}$$

$$J0 = -(\text{Cylinder power}/2) \cdot \cos(2 \cdot \text{Cylinder axis}), \tag{2}$$

$$J45 = -(\text{Cylinder power}/2) \cdot \sin(2 \cdot \text{Cylinder axis}). \tag{3}$$

2.3.2. Height Percentile

From the height and age data (in months) and based on the sex of the participants, the corresponding percentile was calculated according to the LMS method of the Centers for Disease Control and Prevention (CDC) [33]. This consists of applying the Box–Cox transformation (L), the median (M), and the generalized coefficient of variation (S) in the following Equation (4) to obtain the Z-score:

$$Z\text{-score} = [(\text{Body Height}/M)^L - 1]/(L \cdot S), \tag{4}$$

The calculation was performed automatically using the MSD manual clinical height percentile calculator, which can be found at the medical height-for-age percentile calculator for boys [34] and the medical height-for-age percentile calculator for girls [35].

### 2.4. Statistical Analysis

Data analysis was performed with IBM SPSS Statistics v.28.0 for Windows (SPSS Inc., Chicago, IL, USA). The level of significance was set at $p \leq 0.05$ for all analyses. The normality of the study variables was tested using the Kolmogorov–Smirnov test prior to performing the analyses [36]. All variables showed non-normal distribution (Kolmogorov–Smirnov test, all $p \leq 0.043$); therefore, nonparametric tests were conducted. Correlations between refractive parameters, AL, body height, and age, and between refractive parameters, AL, and age based on sex, were assessed by Spearman's correlation test. Correlations were classified as weak (0.20 to 0.40), moderate (0.41 to 0.60), good (0.61 to 0.80), or strong (0.81 to 1.00) [37]. Differences were analyzed using the Kruskal–Wallis or Mann–Whitney U test based on the number of samples. The pairwise comparisons in Kruskal–Wallis analyses were performed using Dunn's post hoc test with Bonferroni correction [38]. Differences in body height and SE were assessed by grouping based on AL, and differences in AL and SE were assessed by grouping based on height percentile. To group the eyes based on AL,

three groups were defined: group 1 consisted of eyes under 23.00 mm in length, group 2 consisted of eyes between 23.00 and 25.00 mm, and group 3 consisted of eyes with over 25.00 mm [39]. Three groups were also defined in the analysis based on height percentile: group 1 comprised participants below the 25th percentile, group 2 comprised participants between the 25th and 75th percentile, and group 3 comprised participants above the 75th percentile [40,41].

## 3. Results

A total of 72 eyes of 36 myopic volunteer participants (19 girls and 17 boys) with a mean age of $11.1 \pm 1.42$ years ranging from 8 to 14 years were included in the study. Descriptive statistics of the sample are shown in Table 1.

**Table 1.** Descriptive statistics of the participants included in the study.

| Parameter | Median [IQR] | Range |
|---|---|---|
| Sphere power (D) | −1.68 [2.10] | −4.50 to −0.12 |
| Cylindrical power (D) | −0.37 [0.47] | −1.50 to 0.00 |
| SE (D) | −1.84 [2.17] | −4.69 to −0.12 |
| J0 vector | 0.00 [0.20] | −0.75 to 0.62 |
| J45 vector | 0.00 [0.20] | −0.43 to 0.68 |
| AL (mm) | 23.86 [1.01] | 22.83 to 26.53 |
| Body height (cm) | 146.00 [15.75] | 125 to 175 |
| Height percentile | 53.18 [41.39] | 4.46 to 99.04 |

AL = Axial Length; IQR = Interquartile Range; SE = Spherical Equivalent.

### 3.1. Correlations between Refractive Parameters, AL, Body Height, and Age

AL was negatively and moderately correlated with sphere power and SE (Spearman's correlation, $r \leq -0.571$, both $p < 0.001$). No significant correlations were found with other refractive components (Spearman's correlation, all $p \geq 0.208$). AL was also positively and weakly correlated with body height (Spearman's correlation, $r = 0.283$, $p = 0.016$) (Table 2). Regarding body height, it was correlated negatively and weakly with sphere power, SE, and J0 (Spearman's correlation, $r \leq -0.315$, all $p \leq 0.019$). No additional significant correlations were found between body height and the other refractive components (Spearman's correlation, all $p \geq 0.188$) (Table 2). Finally, age was strongly correlated with body height (Spearman's correlation, $r = 0.839$, $p < 0.001$) but not with refractive parameters or AL (Spearman's correlation, all $p \geq 0.223$) (Table 2).

**Table 2.** Correlation between refractive parameters, AL, body height, and age.

| | | Sph Power | Cyl Power | Cyl Axis | SE | J0 | J45 | AL | Height |
|---|---|---|---|---|---|---|---|---|---|
| AL | $r_s$ | −0.558 * | −0.150 | 0.119 | −0.571 * | −0.138 | −0.080 | | |
| | $p$ | <0.001 | 0.208 | 0.321 | <0.001 | 0.248 | 0.506 | | |
| Height | $r_s$ | −0.303 * | −0.157 | −0.004 | −0.315 * | −0.276 * | 0.012 | 0.283 * | |
| | $p$ | 0.010 | 0.188 | 0.973 | 0.007 | 0.019 | 0.923 | 0.016 | |
| Age | $r_s$ | −0.134 | −0.096 | 0.008 | −0.145 | −0.219 | 0.008 | 0.143 | 0.839 * |
| | $p$ | 0.260 | 0.423 | 0.949 | 0.223 | 0.065 | 0.950 | 0.231 | <0.001 |

AL = Axial Length; Cyl Axis = Cylinder Axis; Cyl Power = Cylinder Power; SE = Spherical Equivalent; Sph Power = Sphere Power; $r_s$ = Spearman Correlation. * Statistically significant.

### 3.2. Correlations between Refractive Parameters, AL, and Age Based on Sex

There was a negative and weak correlation between age and SE in the female participants (Spearman's correlation, $r = -0.361$, $p = 0.026$), whereas no significant correlation was found between these variables in the male participants (Spearman's correlation, $p = 0.337$). Additionally, there were no significant correlations between AL and age in either the female or male participants (Spearman's correlation, both $p \geq 0.223$).

### 3.3. Differences in Body Height and SE Based on AL

Statistically significant differences in body height between groups were found when dividing participants based on the AL (Kruskal–Wallis test, H = 27.475, $p < 0.001$). The pairwise comparisons showed that the differences were significant between group 1 and group 3 and between group 2 and group 3 (Dunn–Bonferroni post hoc, both $p \leq 0.002$). No significant differences were found between group 1 and group 2 (Dunn–Bonferroni post hoc, $p = 1.000$) (Figure 1a).

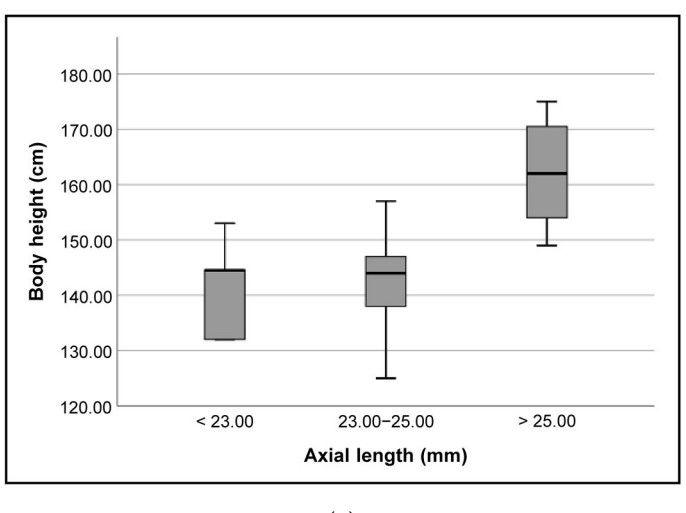
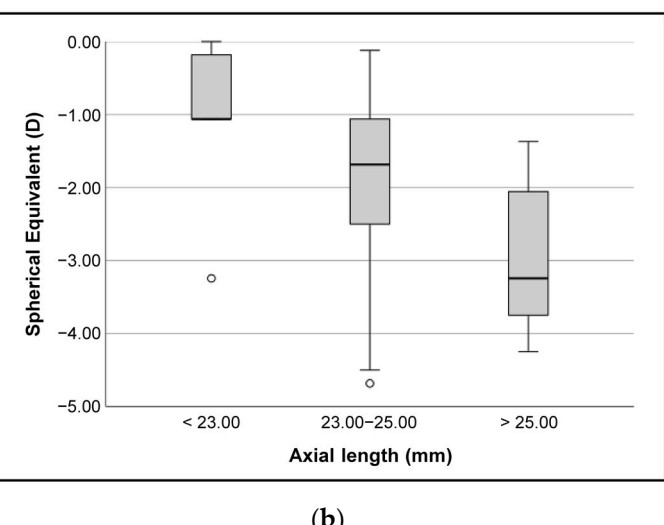

(**a**)                                                                                      (**b**)

**Figure 1.** Variations in morphological parameters as a function of sample AL by Kruskal–Wallis test with multiple comparisons. *n* = 72. (**a**) Differences in body height based on AL. (**b**) Differences in refractive error, described as SE, based on AL. AL = Axial Length; SE = Spherical Equivalent.

The analysis of the differences in SE when grouping participants according to AL found statistically significant differences between the groups (Kruskal–Wallis test, H = 10.381, $p = 0.006$). In the pairwise analysis, differences were again reported in groups 1 and 2 against group 3 (Dunn–Bonferroni post hoc, both $p \leq 0.036$), whereas no significant differences were found between group 1 and group 2 (Dunn–Bonferroni post hoc, $p = 0.290$) (Figure 1b).

For both body height and SE, it was therefore observed that the higher AL group was consistently different from the medium or low AL groups. However, between the medium and low AL groups, there was no difference in either case.

### 3.4. Differences in AL and SE Based on Body Height Percentile

AL exhibited significant differences when dividing participants into three groups based on body height percentile (Kruskal–Wallis test, H = 18.130, $p < 0.001$). Pairwise analysis showed significant differences between group 1 and group 3 and between group 2 and group 3 (Dunn–Bonferroni post hoc, both $p \leq 0.003$). No significant differences were found between group 1 and group 2 (Dunn–Bonferroni post hoc, $p = 0.199$) (Figure 2a).

Significant differences in SE based on body height percentiles were also found (Kruskal–Wallis test, H = 11.554, $p = 0.003$). The pairwise comparison revealed significant differences between group 1 and group 3 and between group 1 and group 2 (Dunn–Bonferroni post hoc, both $p \leq 0.005$). In contrast to previous analyses, no differences were observed between group 2 and group 3 (Dunn–Bonferroni post hoc, $p = 1.000$) (Figure 2b).

Although there was a significant correlation between J0 and body height, the distribution of J0 was consistent across all three height percentile groups (Kruskal–Wallis test, $p = 0.117$).

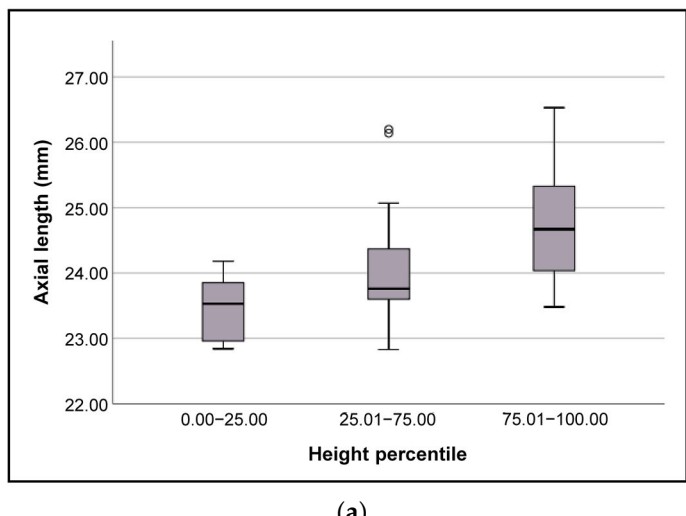 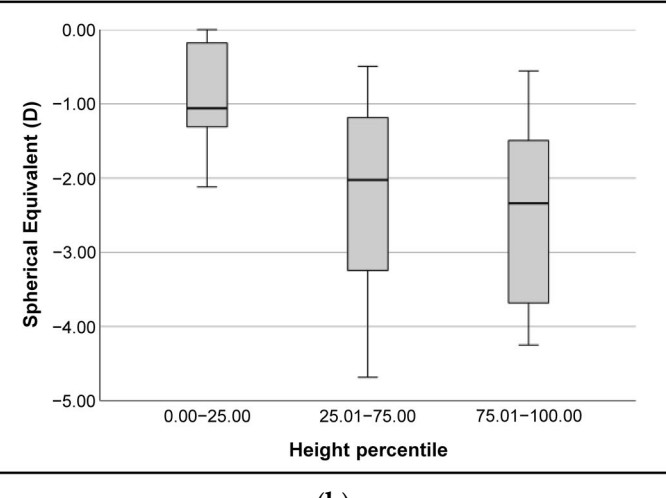

(**a**)                                                                                     (**b**)

**Figure 2.** Variation in ocular parameters as a function of sample body height percentile by Kruskal–Wallis test with multiple comparisons. *n* = 72. (**a**) Differences in AL based on body height percentile. (**b**) Differences in refractive error, described as SE, based on body height percentile. AL = Axial Length; SE = Spherical Equivalent.

### 3.5. Differences in Body Height, AL, and SE Based on Sex

Upon analysis of the data, significant differences were found between the male and female participants for all variables studied: body height (Mann–Whitney U test, U = 462.000, $p$ = 0.038) (Figure 3a), AL (Mann–Whitney U test, U = 377.000, $p$ = 0.002) (Figure 3b), and SE (Mann–Whitney U test, U = 460.500, $p$ = 0.036) (Figure 3c). The boys had higher values than girls for body height (mean rank of 41.91 and 31.66, respectively), AL (mean rank of 29.42 and 44.41, respectively), and SE (mean rank of 41.38 and 31.04, respectively).

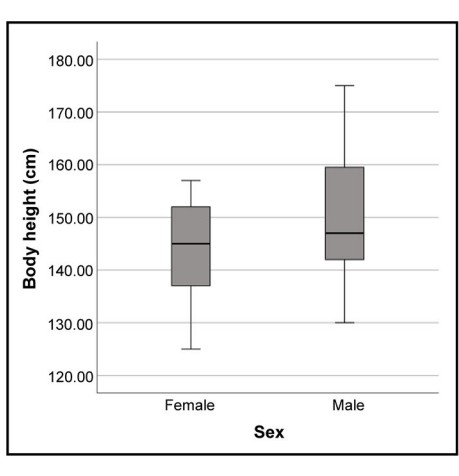 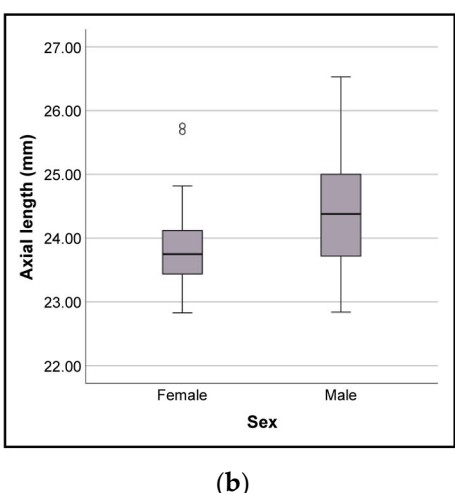 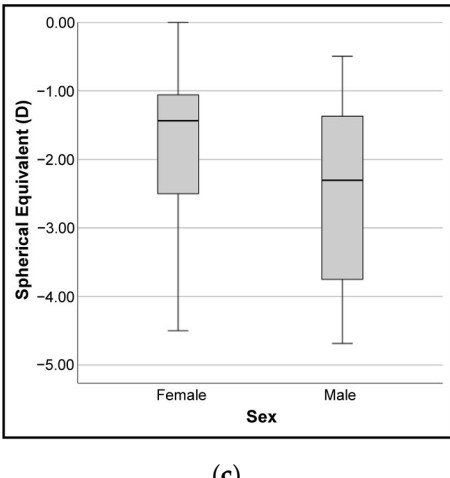

(**a**)                                           (**b**)                                           (**c**)

**Figure 3.** Variation in ocular parameters as a function of sex by Mann–Whitney U test. *n* = 72. (**a**) Differences in body height based on sex. (**b**) Differences in AL based on sex. (**c**) Differences in refractive error, described as SE, based on sex. AL = Axial Length; SE = Spherical Equivalent.

## 4. Discussion

The study results revealed significant correlations of varying strength between AL, body height, and refractive error. The taller individuals had longer eyes, and both factors were associated with more negative refraction. These findings are consistent with prior research [2,42–45]. Concerning the relationships between age and refractive error according to sex, it was observed that for the female participants, older age was associated with

more negative refractive error. However, this was not found in the male participants. The discrepancy may be influenced by the different timing of puberty in girls and boys [46] and may not be observed in other age groups [20].

Regarding the differences, it was found that body height is significantly higher in children with an AL of more than 25.00 mm compared to children with shorter AL. This indicates the possibility of genetic factors affecting both characteristics simultaneously or nutritional factors related to overall growth [47]. On the other hand, the lack of difference between groups 1 and 2 generates some debate as it is possible that the variability in AL is not sufficient to generate statistical differences, for example, due to the existence of a specific undetected threshold. It may also mean that the association between AL and body height is nonlinear, being not constant over the entire range of AL. In order to refocus this analysis, children were divided into three groups based on their body height percentile and differences in AL were then assessed. The results were similar to those mentioned above. Differences were found between the highest percentile group and the other two groups but not between groups 1 and 2. This may be associated with a more pronounced overall growth in that height category, or, as noted previously, it is possible that there is an undetected threshold between groups 1 and 2. Previous studies have also found that subjects with longer AL were taller than subjects with shorter eyes [42–45].

In addition to the interactions between AL and body height, the interaction of refractive error with these parameters was also examined. As occurred between body height and AL, the SE was also significantly more negative in the higher AL group than in the two lower groups, but there was no difference between groups 1 and 2. This supports the hypothesis of a nonlinear relationship. As AL increases, the SE becomes more negative, but not in a proportional magnitude. Compensatory mechanisms during the emmetropization process may be partly responsible for this disparity in differences. In myopic eyes, there is a tendency for the cornea to flatten in compensation [48,49]. Previous studies have shown a relationship between corneal curvature and AL, but this relationship is not linearly related to the refractive error [50]. When the differences in SE were evaluated based on body height percentile, there was a change from the trends of previous analyses. Differences appear with the lowest height group, with no significant differences between group 2 and group 3. Previous studies have also identified interactions between SE and AL or body height. These reported that the SE was more negative at higher values of AL and body height [2,43].

Finally, in comparisons based on sex, a greater magnitude of body height, AL, and SE was found in the boys. The findings are in line with previous research [14,15,42–44,51]. However, there are also reports of the opposite effect, stating that girls were more likely to develop high myopia [14,52] or that their prevalence of myopia was higher than that of boys [15]. It has been suggested that this sex difference in body height is responsible for the greater AL of the boys [53], but Hashemi H et al. [13] and Okabe et al. [43] found that the differences in AL remain even after adjusting for height. Consistent with the longer AL in the boys, their refractive errors were significantly more negative compared to those of the girls. Rudnicka et al. [15] suggest that the disagreement between studies may be attributed to the age of the children studied and the statistical power of each study. In the present study, the sample design was planned on the assumption that the myopia progression experiences its primary increase between the ages of 8 and 15 years old [24]. This timeframe aligns with official growth data from the CDC [33], which indicates a more significant increase in body height during this period. It is possible that differences based on sex would not have been found if a younger population had been analyzed.

Previous studies have proposed an alternative approach based on formulation to estimate AL either from refractive error alone or from a combination of refractive error and corneal curvature [19]. Following this concept, the present results suggest the hypothesis that employing a mathematical approach may incorporate height or even sex into the equation. This approach should be calculated in a cross-sectional study and subsequently tested in a longitudinal sample, similar to previous designs [20]. Additionally, beyond simply optimizing theoretical equations for estimating AL by incorporating the studied parameters

here, enhancing the formulas used in myopia progression predictor calculators integrated into clinical practice (whether in online or app calculators and the latest-generation devices equipped with predictive software) may improve professional management and ensure more accurate communication with patients.

The present study had some limitations. Firstly, the sample consisted only of children aged 8 to 14, potentially restricting the generalization of the results to older age groups; a larger and more diverse sample would improve the applicability of the findings to the general population, given the variation in myopia across different developmental stages. In addition, subsequent studies might consider including a follow-up period to examine potential changes in the relationship between the values of the studied parameters during the progression or control of myopia over an extended period.

In summary, there is a relationship between the AL of the eye and its refractive status. In addition, individual anatomical characteristics such as body height and sex also play a role in these parameters. It was observed that AL and myopic refractive error are higher in the taller and male subjects.

**Author Contributions:** Conceptualization, V.N.-P., N.N.-P., J.G.-Q., M.J.G., H.P.-V. and E.Y.-P.; Data curation, V.N.-P., N.N.-P. and J.G.-Q.; Formal analysis, V.N.-P., N.N.-P. and J.G.-Q.; Funding acquisition, H.P.-V. and E.Y.-P.; Investigation, V.N.-P. and N.N.-P.; Methodology, V.N.-P., N.N.-P. and H.P.-V.; Project administration, M.J.G., H.P.-V. and E.Y.-P.; Resources, V.N.-P.; Software, V.N.-P., J.G.-Q. and H.P.-V.; Supervision, M.J.G., H.P.-V. and E.Y.-P.; Validation, V.N.-P., N.N.-P., J.G.-Q. and H.P.-V.; Visualization, V.N.-P., J.G.-Q., H.P.-V. and E.Y.-P.; Writing—original draft, V.N.-P., N.N.-P. and H.P.-V.; Writing—review and editing, V.N.-P., J.G.-Q., M.J.G., H.P.-V. and E.Y.-P. All authors have read and agreed to the published version of the manuscript.

**Funding:** This research did not receive any specific grant or financial support from funding agencies in the public, commercial, or not-for-profit sectors. The study was conducted independently, and the authors did not have any external financial assistance or sponsorship for this investigation.

**Institutional Review Board Statement:** The study was conducted in accordance with the Declaration of Helsinki and approved by the Bioethics Committee of the Universidade de Santiago de Compostela (Approval code: USC 04/2022. Date of approval: 25 February 2022).

**Informed Consent Statement:** Written informed consent was obtained from the legal guardian of each participant involved in the study.

**Data Availability Statement:** Data is unavailable due to privacy restrictions.

**Conflicts of Interest:** The authors declare no conflicts of interest.

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
