# Peer review of "Associations between Ocular Biometry, Refractive Error, and Body Characteristics"

_photonics, doi:10.3390/photonics11020165_

Round 1

Reviewer 1 Report

Comments and Suggestions for Authors

This paper discussed the association between AL, height and SE of young age people. This paper is well structured, and statistical analyses are well written. But There are vague points to consider.

Specific comments:

1. what is the purpose of this research in the view of usefulness. You must improve your manuscript in this point. 

-> For example, your research can be used in specific treatment or development of medical instrument. 

-> AL can be measured directly by a Topcon MYAH biometer. It is needless to estimate the AL from height or SE.

2. In my thinking, AL can be regressed by using deep learning system of face picture or others. Recent progress of vision technology must be contained in this manuscript for easy reading. The link between your research and deep neural network can be discussed for clear reading. 

Comments on the Quality of English Language

I could find small grammatical errors in line 45 and line 89.

Author Response

DETAILED RESPONSES TO REVIEWER 1

Comments and Suggestions for Authors

This paper discussed the association between AL, height and SE of young age people. This paper is well structured, and statistical analyses are well written. But There are vague points to consider.

- Response: The authors would like to thank the reviewers for their suggestions to improve the manuscript. The authors believe that the comments have identified important areas that require clarification. Following the reviewer's indications, the rationale and significance behind has been clarified, along with new references to justify the study.

Specific comments:

  1. what is the purpose of this research in the view of usefulness. You must improve your manuscript in this point.

-> For example, your research can be used in specific treatment or development of medical instrument.

-> AL can be measured directly by a Topcon MYAH biometer. It is needless to estimate the AL from height or SE.

- Response: Thank you for the suggestion. Following the reviewer's advice, a paragraph has been added to the discussion section. Based on a previous hypothesis [1], a proposal for a new mathematical approach for indirect estimation of AL, incorporating body height and sex, may be derived from the present findings. Knowing the relationships between AL growth and other individual morphological characteristics, such as body height, is a small step towards understanding the causes and progression of myopia in children. Additionally, beyond just optimizing equations for estimating AL, the findings of this study may also contribute to enhancing the formulas used in myopia progression predictor calculators that are currently incorporated into clinical practice, both in online or app calculators and in the latest-generation devices available in the market equipped with predictive software. This integration not only enhances professional management but also ensures accurate communication with patients.

[1] Philip B Morgan, Sara J McCullough, Kathryn J Saunders. Estimation of ocular axial length from conventional optometric measures Cont Lens Anterior Eye. 2020. 43(1):18-20.

I could find small grammatical errors in line 45 and line 89.

- Response: Apologies regarding the typos. The typographical errors have been corrected in the text.

Reviewer 2 Report

Comments and Suggestions for Authors

Padin V, N., et al elucidate the impact of myopia, a prevalent refractive error globally associated with excessive axial length (AL) of the eye, which can lead to serious consequences, including blindness. The authors assess the connections between AL, body height, refractive error, and gender, involving 36 myopic participants aged 11.1 ± 1.42 years, revealing significant correlations among AL, body height, and spherical equivalent (SE). Moreover, grouping participants by AL or height percentile unveils noteworthy differences in body height and SE.

According to the authors, the novelty of this study is that gender differences are observed in SE, AL, and body height, highlighting the intricate relationship between AL and individual anatomical characteristics influencing the eye's refractive state. The authors examined the Individual anatomical attributes, exemplified by factors like body height and gender, leading to an empirical observation discerned that taller and male subjects exhibit elevated values in axial length (AL) and myopic refractive error.

The objectives of this study carry significant import; however, a pivotal lacuna exists concerning the escalating prevalence of aging and heightened refractive errors, as highlighted by Fricke et al. (2018). Although the authors mentioned certain constraints, among them most crucial is the sample exclusively comprises children aged 8 to 14, posing a potential limitation in extrapolating the findings to older age cohorts.

The experiments designed for this study are justified and the results are significant. The introduction and the discussion were written clearly with proper information and references. High appreciation should go to the authors as they mention ethics and patient consent in the methods.

Nonetheless, the article seemed to possess good value toward the relationship between the AL of the eye and its refractive status depending on parameters like body height and sex.

Overall, the clarity of the text needs very few readjustments. The manuscript has minor typographical and grammatical errors. The results and the figures were consistent based on the written legends and results. The quantitative analyses are much appreciated but need more careful revision based on the significance of the data. In general, the manuscript can accomplish the caliber of quality for consideration for publication in the Journal of “Photonics” with minor changes. The authors are advised to consider the comments below:

Comments:

1.      Could you please provide a brief explanation what would be the correlation of “aging and increasing refractive errors” based on the sex?

2.      What would be the speculation if you consider a broader population with different age groups, considering the variable nature of myopia across distinct developmental stages?

3.      Could you please provide a graph showing differences in body height, AL, and SE based on sex?

4.      Please mention the sign – on top of the error bars in Figure 2- graph (a).

Comments on the Quality of English Language

The manuscript has minor typographical and grammatical errors. 

Author Response

DETAILED RESPONSES TO REVIEWER 2

Comments and Suggestions for Authors

Padin V, N., et al elucidate the impact of myopia, a prevalent refractive error globally associated with excessive axial length (AL) of the eye, which can lead to serious consequences, including blindness. The authors assess the connections between AL, body height, refractive error, and gender, involving 36 myopic participants aged 11.1 ± 1.42 years, revealing significant correlations among AL, body height, and spherical equivalent (SE). Moreover, grouping participants by AL or height percentile unveils noteworthy differences in body height and SE.

According to the authors, the novelty of this study is that gender differences are observed in SE, AL, and body height, highlighting the intricate relationship between AL and individual anatomical characteristics influencing the eye's refractive state. The authors examined the Individual anatomical attributes, exemplified by factors like body height and gender, leading to an empirical observation discerned that taller and male subjects exhibit elevated values in axial length (AL) and myopic refractive error.

The objectives of this study carry significant import; however, a pivotal lacuna exists concerning the escalating prevalence of aging and heightened refractive errors, as highlighted by Fricke et al. (2018). Although the authors mentioned certain constraints, among them most crucial is the sample exclusively comprises children aged 8 to 14, posing a potential limitation in extrapolating the findings to older age cohorts.

The experiments designed for this study are justified and the results are significant. The introduction and the discussion were written clearly with proper information and references. High appreciation should go to the authors as they mention ethics and patient consent in the methods.

Nonetheless, the article seemed to possess good value toward the relationship between the AL of the eye and its refractive status depending on parameters like body height and sex.

Overall, the clarity of the text needs very few readjustments. The manuscript has minor typographical and grammatical errors. The results and the figures were consistent based on the written legends and results. The quantitative analyses are much appreciated but need more careful revision based on the significance of the data. In general, the manuscript can accomplish the caliber of quality for consideration for publication in the Journal of “Photonics” with minor changes. The authors are advised to consider the comments below:

- Response: The authors would like to thank the reviewer for their suggestions and kind words about the manuscript. Following the reviewers' indications, the entire manuscript has been re-edited to improve readability and clarify any unclear points. As the reviewer suggests, some changes were made in all sections, with special attention to the rationale behind the study and also in the sample design, adding further information regarding the limited age range choice in the discussion section.

Comments:

  1. Could you please provide a brief explanation what would be the correlation of “aging and increasing refractive errors” based on the sex?

- Response: Thank you for the suggestion. Based on the reviewer's recommendations, an additional analysis of correlations between age, refractive parameters, AL and height has been added in section 3.1 of the manuscript. In addition, for a better visualisation of the data, table 2 has been extended to include it. Finally, comments about those new analyses and findings were added to the discussion section.

  1. What would be the speculation if you consider a broader population with different age groups, considering the variable nature of myopia across distinct developmental stages?

- Response: Thank you for the suggestion. In the present study, the sample design was planned based on the hypothesis that myopia progression experiences its primary increase between the ages of 8 and 15 years old, after which it begins to slow down [1]. This timeframe aligns with data obtained from the official growth chart of the Centers for Disease Control and Prevention (https://www.cdc.gov/growthcharts/clinical_charts.htm), which indicates that the most significant stature growth occurs during this period. Based on the input provided by the reviewer, the rationale here exposed for the sample design was added to the manuscript in the discussion section.

[1] Cooper, J; Tkatchenko, A.V. A Review of Current Concepts of the Etiology and Treatment of Myopia. Eye Contact Lens 2018, 44, 231-247

  1. Could you please provide a graph showing differences in body height, AL, and SE based on sex?

- Response: Thank you for the suggestion. A graph has been included for each of the three cases (see Figure 3).

  1. Please mention the sign – on top of the error bars in Figure 2- graph (a).

- Response: Thank you for the suggestion. The symbols depicted atop the error bars in Figure 2 serve to indicate values that extend beyond the confines of the bar chart. A similar visualization approach is employed in Figure 1b, where specific values exhibit a greater SE than their respective group averages. The same situation is reflected in Figure 3b. If the reviewer considers this to be confusing, the authors are committed to exploring alternative methods of representation in further revisions.

Reviewer 3 Report

Comments and Suggestions for Authors

I have had the opportunity to review your manuscript titled  "Associations between Ocular Biometry, Refractive Error, and Body Characteristics"  submitted to the Journal of Photonics for the section on Latest Developments in Ocular Biometry. While the topic you have chosen is of significant interest in the field, I would like to provide some constructive feedback to enhance the quality and impact of your research.

It is crucial to acknowledge that there are numerous existing studies in this domain, many of which involve extensive participant numbers and longitudinal data. These studies have already laid a substantial foundation for understanding the associations in ocular biometry. To make a meaningful contribution, your study needs to delineate how it adds to or differs from these existing bodies of work.

A fundamental issue with your study is the relatively small sample size. Given the breadth of existing research with larger cohorts, your study may not provide sufficient statistical power to draw significant conclusions or to make a novel contribution to the field. Consider either increasing your sample size or focusing on a niche aspect not extensively covered in existing literature.

The exclusion of critical variables such as age and race limits the depth of your analysis. These factors are vital for a nuanced understanding of ocular biometry and should be included in a more comprehensive study. Their inclusion would also facilitate a more meaningful comparison with other extensive research in the field.

The current presentation of your data, particularly in the plots, lacks the clarity and depth required for a publication of this nature. The plots would benefit from including a broader range of correlations, such as between age, height, and axial length, rather than the narrower focus currently presented.

Your manuscript mentions an analysis of male and female participants but fails to visually present this data. Comparative plots highlighting any gender-based differences would significantly enhance the understanding and impact of your findings.

Given the extensive existing research in this area, your study needs to clearly demonstrate innovation or a novel approach to be considered for publication. This could involve a unique methodology, a focus on an under-researched demographic, or novel insights that challenge or extend current understanding.

Finally, the conclusions drawn in your study need to be supported by more robust evidence. Given the extensive prior research in this area, any claims made need to be firmly established and backed by solid data.

In conclusion, while your study addresses a topic of importance, it requires significant revision to stand out in the crowded field of ocular biometry research. By addressing these concerns, I believe your study can make a valuable contribution to the scientific community.

Author Response

DETAILED RESPONSES TO REVIEWER 3

Comments and Suggestions for Authors

I have had the opportunity to review your manuscript titled "Associations between Ocular Biometry, Refractive Error, and Body Characteristics" submitted to the Journal of Photonics for the section on Latest Developments in Ocular Biometry. While the topic you have chosen is of significant interest in the field, I would like to provide some constructive feedback to enhance the quality and impact of your research.

It is crucial to acknowledge that there are numerous existing studies in this domain, many of which involve extensive participant numbers and longitudinal data. These studies have already laid a substantial foundation for understanding the associations in ocular biometry. To make a meaningful contribution, your study needs to delineate how it adds to or differs from these existing bodies of work.

A fundamental issue with your study is the relatively small sample size. Given the breadth of existing research with larger cohorts, your study may not provide sufficient statistical power to draw significant conclusions or to make a novel contribution to the field. Consider either increasing your sample size or focusing on a niche aspect not extensively covered in existing literature.

The exclusion of critical variables such as age and race limits the depth of your analysis. These factors are vital for a nuanced understanding of ocular biometry and should be included in a more comprehensive study. Their inclusion would also facilitate a more meaningful comparison with other extensive research in the field.

The current presentation of your data, particularly in the plots, lacks the clarity and depth required for a publication of this nature. The plots would benefit from including a broader range of correlations, such as between age, height, and axial length, rather than the narrower focus currently presented.

Your manuscript mentions an analysis of male and female participants but fails to visually present this data. Comparative plots highlighting any gender-based differences would significantly enhance the understanding and impact of your findings.

Given the extensive existing research in this area, your study needs to clearly demonstrate innovation or a novel approach to be considered for publication. This could involve a unique methodology, a focus on an under-researched demographic, or novel insights that challenge or extend current understanding.

Finally, the conclusions drawn in your study need to be supported by more robust evidence. Given the extensive prior research in this area, any claims made need to be firmly established and backed by solid data.

In conclusion, while your study addresses a topic of importance, it requires significant revision to stand out in the crowded field of ocular biometry research. By addressing these concerns, I believe your study can make a valuable contribution to the scientific community.

- Response: The authors would like to thank the reviewers for their detailed comments and suggestions about the manuscript. The authors believe that the comments have identified important areas that require improvement. Following the reviewers' indications, the entire manuscript has been re-edited to improve readability and clarify any unclear points. As the reviewer suggests, some changes were made in all sections. The authors agree with the referee on the need for additional analysis to explain the relevance of the study, as well as the need for more context at some points of the manuscript to make a rational explanation for the study performance and results

In terms of the novelty of the present study, gender differences in SE, AL and body height are observed, highlighting the intricate relationship between AL and individual anatomical characteristics that influence the refractive status of the eye. The results may be of interest in optimizing the mathematical equations developed to predict AL when it cannot be measured with appropriate equipment. This information has been presented in the Introduction section. Following the reviewer's indications, information regarding the sample calculation has also been added to clarify the reliability of the results. Regarding exclusion criteria, race was not used as a screening variable in the present study, but age was. Only children between 8 and 14 years of age were included. This narrow range was chosen based on previous studies that have found that myopia progression experiences its primary increase in this age range. Information on this has been added in the discussion section As for the suggested graphs, new graphs have been added for the sex-based analyses. In accordance with the reviewer's suggestions, information on correlations with age has been included, but no graphs have been added for their representation. This is motivated by the fact that a total of 21 graphs would be needed, or 7 if only statistically significant correlations were included. However, if the reviewer deems that they should still be included, we will add them in the next revision.

Round 2

Reviewer 1 Report

Comments and Suggestions for Authors

Authors responded to all my comments. Although the usefulness still is in incomplete points, authors  sincerely improved the manuscript in this point according to my comments. 

Reviewer 3 Report

Comments and Suggestions for Authors

I have carefully reviewed the revised manuscript titled "Associations between Ocular Biometry, Refractive Error, and Body Characteristics," submitted to the Journal of Photonics. The modifications made have effectively addressed the concerns and comments raised during the initial review, leading to improved clarity and readability. The inclusion of additional analyses and contextual information enhances the overall quality of the study.

I acknowledge the clarification regarding the exclusion criteria, specifically the narrow age range of participants between 8 and 14 years. This information has been added to the discussion section, providing a rationale for the age selection based on the observed increase in myopia progression during this specific age range.

The addition of new graphs for sex-based analyses is a positive step, contributing to a more comprehensive understanding of the study's findings. I appreciate the consideration given to including information on correlations with age, as suggested. While I understand the challenges in presenting a large number of graphs, I encourage you to include relevant visual representations if they significantly contribute to the clarity and depth of the analysis. Overall, the efforts made in response to my comments have substantially improved the manuscript.